# Merged Multi-Sensor Ocean Colour Chlorophyll Product Evaluation for the British Columbia Coast

**Sejal Pramlall [1,*], Jennifer M. Jackson [2,3], Marta Konik [1,4] and Maycira Costa [1]**

[1]   Spectral Lab, Department of Geography, University of Victoria, Victoria, BC V8P 5C2, Canada
[2]   Hakai Institute, Victoria, BC V8W 1T4, Canada
[3]   Fisheries and Oceans Canada (DFO), Sidney, BC V8L 5T5, Canada
[4]   The Institute of Oceanology, Polish Academy of Sciences, 81-712 Sopot, Poland
[*]   Correspondence: pramlall.sejal@gmail.com

**Abstract:** Phytoplankton phenology studies require a dataset that is continuous in time and space since missing data have been shown to affect the accuracy of seasonality metrics. The interpolated GlobColour product provided by the Copernicus Marine Environment Monitoring Service (CMEMS) meets these requirements by being 'gap filled', thus yielding the highest spatial coverage. Despite being validated on a global scale, a regional comparison to in situ Chl-a concentrations should be conducted to enable product application in optically complex waters. This study aims to evaluate the performance of the GlobColour interpolated product in British Columbia coastal waters via a statistical match-up analysis and a qualitative analysis to determine whether the data reflect the region's large-scale seasonal trends and latitudinal dynamics. Additionally, the statistical performance of the GlobColour interpolated product was compared to the original GlobColour and Ocean Colour Climate Change Initiative (OC-CCI) merged chlorophyll-a products based on in situ observations. The GlobColour interpolated product performed relatively well and was comparable to the best-performing product for each water type (RMSE = 0.28, $r^2$ = 0.77, MdAD = 1.5, BIAS = 0.90). The statistics for all the products degraded in Case 2 waters, thus highlighting the dilemma of applying algorithms designed for Case 1 waters in Case 2 waters. Our results indicate how the quality of products can vary in different environmental conditions.

**Keywords:** GlobColour; OC-CCI; chlorophyll-a; match-up; merged products

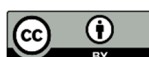

## 1. Introduction

Ocean-colour-derived chlorophyll-a concentration (Chl-a), interpreted as a proxy for phytoplankton biomass, is one of 54 essential climate variables (ECVs) that critically contribute to the characterization of Earth's climate [1]. In order to provide the empirical evidence needed to understand and predict the evolution of climate change, ECVs need detailed historical records spanning decades at a global scale to be able to discern natural changes from anthropogenically induced change [2]. Ocean colour satellites meet these requirements by providing global coverage of Chl-a products at relatively high spatial resolutions (<1 km) and regular sampling frequencies since 1978 [3]. Unfortunately, no single ocean colour satellite encompasses this entire 40-plus year duration, thus merging data from different satellites is needed to generate a continuous time series [4]. The merging process is complex because each sensor possesses distinct features, such as orbits (different timing of overpasses), swaths and revisit times, spatial resolutions (ranging from 300 m–4 km), and spectral resolutions (number and position of spectral wavebands) [4]. This merging process must, therefore, be carefully thought out to prevent biases, artifacts, or discontinuities from being introduced [4,5].

The GlobColour project and the Ocean Colour Climate Change Initiative (OC-CCI) produce merged Chl-a products on a global scale by integrating Chl-a estimates from

multiple sensors to minimise temporal discontinuities and spatial biases among single satellite sensors [6–9]. The GlobColour project was developed in 2005 by the European Space Agency (ESA) as a data user element program to provide a continuous dataset of merged level-3 ocean colour products that accommodate global carbon cycle research (ACRI-ST GlobColour Team, 2020). Subsequently, in 2010 the ESA launched the OC-CCI to produce a coherent, long-term, and error-characterized chlorophyll product to meet the needs and quality standards required for ECVs [7,9]. Despite merged products improving the data coverage in both time and space, missing data due to sun glint, persistent cloud cover, atmospheric aerosols, or sensor saturation over ice, sand, or snow is still prevalent. GlobColour, therefore, provides an additional daily 'cloud free' Chl-a interpolated product that minimises missing data [10].

Missing data are particularly challenging for studies concerning phytoplankton phenology, as they have been shown to affect the accuracy of seasonality metrics [11]. When missing data concur with the timing of bloom initiation, termination, and peak amplitude, the estimations in phenological indices can suffer from inaccuracies and systematic biases in the computation method [12]. It is, therefore, necessary to compensate for the missing data prior to deriving the phenology indices by interpolating across gaps [13]. The only product that provides continuous spatial and temporal data that meets the requirements of phenology studies is the GlobColour interpolated product.

Here, we evaluate whether the GlobColour interpolated product is suitable to use in the waters of British Columbia (B.C.), which consist of both Case 1 and Case 2 waters. Despite these products being validated and assessed for quality standards on a global scale [10,14,15], it is recommended to compare products to in situ water samples [7] since Chl-a products have shown regional accuracy biases when compared to observed data [16]. Specifically on the west coast of Canada, the evaluation of Chl-a product performance is necessary due to the optically complex nature of B.C. waters [17–22]. Coastal waters are under the influence of many riverine systems discharging terrigenous materials, including inorganic sediments and dissolved organic matter, which impact water bio-optical properties [18,22–27]. Off-shelf waters are optically simpler because particles are primarily composed of phytoplankton [28].

As a first step to acquiring a suitable long-term spatially and temporally continuous time series of Chl-a for the B.C. coast, this study evaluates the GlobColour interpolated Chl-a product in relation to in situ Chl-a data, including a comparison with the original GlobColour Chl-a product and OC-CCI. An additional qualitative analysis is performed to determine whether the data reflect the region's large-scale seasonal trends and latitudinal dynamics. By conducting a regional validation of the GlobColour interpolated product, this research shows that the product can be used for monitoring applications, such as phytoplankton phenology and bioregionalization studies, that allow for data incorporation into marine management strategies along the B.C. coast.

## 2. Materials and Methods

### 2.1. Study Area

The study area, extending from 47–60°N and 122–140°W (Figure 1A), included the coastal waters off northern Washington state, British Columbia, southeast Alaska, and the adjacent open ocean waters of the subarctic northeast Pacific Ocean. The continental shelf off B.C. is generally narrower than 45 km, reaching 95 km in the shallow basins of the Hecate Strait and Queen Charlotte Sound [27]. In this region (Figure 1A), phytoplankton productivity varies spatially and interannually. The highest phytoplankton biomass and earliest blooms have been observed in the Strait of Georgia on the southern B.C. coast, with surface concentrations ranging from <1 mg m$^{-3}$ in winter to >15 mg m$^{-3}$ during bloom conditions, and the second-highest biomass along the southwest coast of Vancouver Island [18,29,30]. This high productivity can be attributed to upwelling favourable winds in summer (April–September), which bring nutrient-rich water towards the surface

[29,31–34]. Similarly, freshwater discharge increases both the stability and the supply of land-derived nutrients, which keep both nutrients and phytoplankton in the euphotic zone [29,32,34]. The northern B.C. coast up to the western Alaska regions are dominated by downwelling favourable winds most of the year, though nutrients supplied by river discharge, current-induced upwelling, coastal eddies, and winter upwelling winds increase phytoplankton production [34].

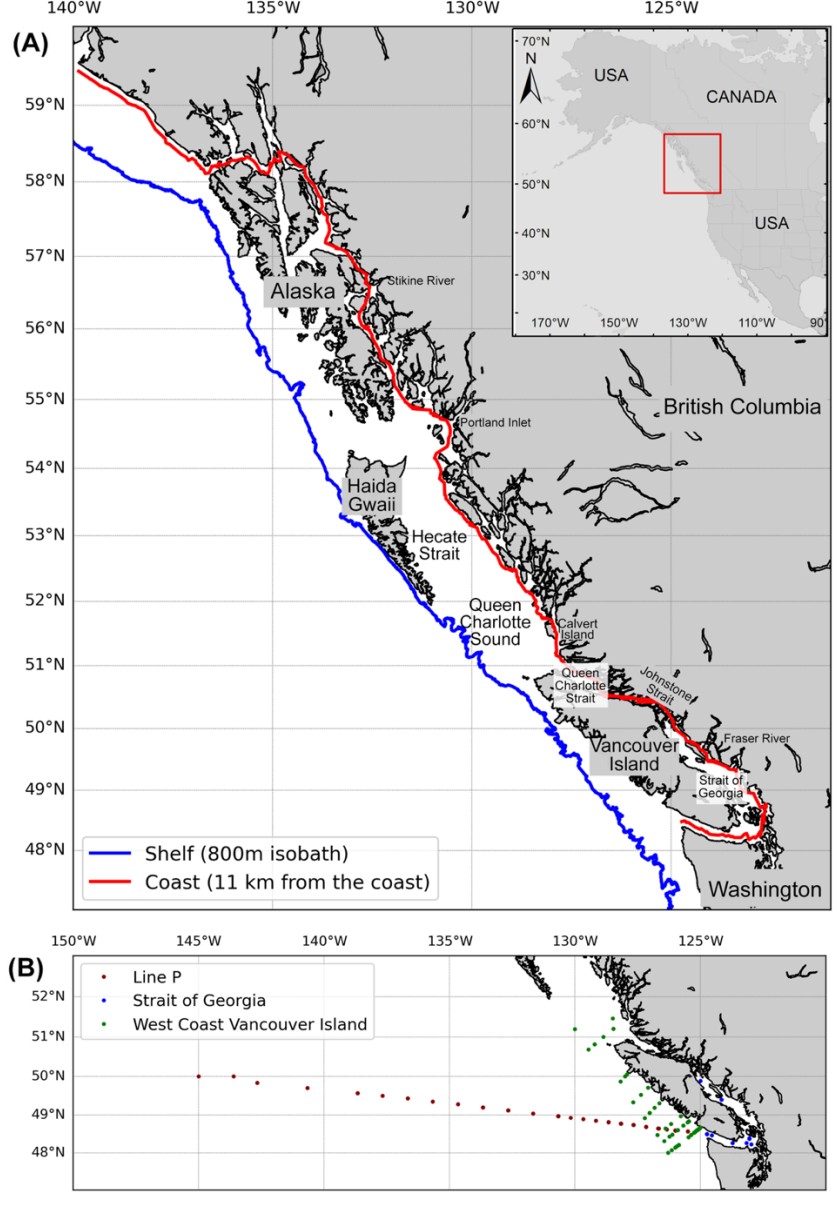

**Figure 1.** (**A**) Study area map with names of the main rivers and locations cited in the text and transect lines used to extract satellite product data from the coast (red line) and continental shelf over the 800 m isobath (blue line). (**B**) Locations of in situ Chl-a data collected by DFO.

### 2.2. Datasets

The performance of the GlobColour interpolated (Chla-GCint) product was evaluated in the coastal and open ocean waters of B.C. using in situ Chl-a data provided by Fisheries and Oceans Canada (DFO) via a one-to-one match-up analysis and compared to the merged multi-sensor satellite Chl-a datasets, GlobColour CHL1 (Chla-GC), and OC-CCI (Chla-OC).

### 2.2.1. In Situ Chl-a Data

Chl-a data were collected by DFO using high-performance liquid chromatography (HPLC) methods as part of their monitoring of the marine ecosystem. The validation dataset (total = 1914) consisted of the northeast Pacific measurements of physical, chemical, and biological data (Figure 1B), which are conducted along Line P three times a year in winter, spring, and summer (2006–2017); off the west coast of Vancouver Island twice a year in spring and summer (2004–2017); and in the Strait of Georgia three times a year in spring, summer, and fall (2011–2017). Briefly, water samples were filtered onto 47 mm GF/F filters and stored at −80 °C prior to analysis. In the lab, samples were extracted in 95% methanol at −20 °C for 24 h and analyzed with a WATERS 2695 HPLC separations system, as detailed in [35]. Only surface samples less than 6.1 m were considered, and values flagged by DFO as poor measurements were removed prior to analysis. Additional information and the datasets can be found at https://open.canada.ca/data/en/dataset/871B0b32-3135-40c8-868e-c5d87800ca76 (accessed on 12 April 2021). Other data sources were not used in an attempt to be consistent with the HPLC analytical methods implemented by DFO.

### 2.2.2. Merged Multi-Sensor Satellite Data

GlobColor and OC-CCI data products used in this study were open-access daily global data at a 4 km × 4 km spatial resolution downloaded for the period of 1998–2021. Due to low solar elevation conditions in this region during the winter months, the analysis was computed using nine months of data, from mid-February to mid-November [17,29,30].

GlobColour Daily Chl-a Product

GlobColour products correspond to continuous 23-year time series (1997-present) created by merging data from the following sensors: SeaWiFS (1997–2010), MERIS (2002–2012), MODIS-Aqua (2002-present), VIIRS-NPP (2012-present), OLCI-S3A (2016-present), VIIRS-SNPP (2017-present), and OLCI-S3B (2019-present). GlobColour provides merged products using three merging techniques: (i) a simple averaging method (AV) of Level 2 Chl-a estimates, (ii) weighted averaging (AVW) of Level 2 Chl-a estimates adjusted to MERIS using the OC4Me algorithm, and (iii) using the Garver–Siegel–Maritorena (GSM) model to merge Level 3 normalized water-leaving radiances across sensors prior to producing Chl-a retrievals [6,7]. The algorithms used to determine chlorophyll concentration also vary. The CHL1 product is computed using classic ratios applicable for Case 1 waters (available in AVW and GSM merging), CHL-OC5 is based on the blended OC5/CI-Hu algorithm (and AVW merging), and to obtain CHL2 neural networks trained for Case 2 waters are used (with AV merging) [6]. Due to the limited temporal availability of CHL2 (2002–2012), it was not considered in this analysis. The initial validation of CHL-OC5 resulted in poor statistics (not shown). Therefore, only the CHL1 algorithm was further considered for this analysis.

The CHL1 product merged using the GSM model has been shown to provide more accurate chlorophyll concentration estimates in the northeast Pacific [20] and the best fit to in situ Chl-a compared to alternative GlobColour products [6,36]. Although the CHL1 algorithm provides the best performance over Case 1 waters and has not been recommended for use over optically complex coastal waters [6,37], the application of Case 1 algorithms to coastal waters can occur, provided that a verification of Chl-a concentration using in situ measurements confirms the reliability of the product [29]. GlobColour data from http://hermes.acri.fr (accessed on 12 April 2021).

GlobColour Interpolated Chl-a Product

Since 2016, the Copernicus Marine Environment Monitoring Service (CMEMS) has provided a 'cloud free' interpolated daily product, significantly improving the quality and

coverage of daily data upstream [10]. This interpolated product is produced by first merging Chl-a fields estimated using different algorithms (OC3 and OC5 for Case 1 and 2 waters, respectively) generated from Level 2 reflectance [38]. This Level 3 multi-algorithm product is then used as input for spatio-temporal interpolation, an advanced version of the standard optimal interpolation technique that includes anisotropic covariance models at the coastline for better reconstruction of coastal gradients [38]. Interpolated GlobColour data from https://resources.marine.copernicus.eu/product-detail/OCEANCOLOUR_GLO_CHL_L4_REP_OBSERVATIONS_009_082/DATA-ACCESS (accessed on 20 October 2021)

OC-CCI Chl-a Product

The OC-CCI Chl-a dataset (v5.0) is generated by shifting the wavelengths of SeaWiFS, MODIS, VIIRS, and OLCI data to match MERIS bands, applying bias correction, merging the datasets, and computing per pixel uncertainty estimates [39]. POLYMER (v.4.12) atmospheric processing is applied, and Chl-a is estimated using a blended algorithm including the OCI, OCI2, OC2, OC3, OCx (the updated OC3/OC4 band ratio algorithm [40]), and OC5 algorithms, which attempt to weight the outputs of the best-performing algorithm based on the water types present [39]. The OC-CCI dataset, Version [v5.0], ESA, from http://www.esa-oceancolour-cci.org/ (accessed on 21 September 2021).

*2.3. Satellite Data Analysis*

To ensure that the Chl-a concentrations derived from the merged satellite products were accurate in B.C. waters, the Chl-a retrievals were compared to the in situ measurements. Match-up product Chl-a data were obtained from the pixel (16 km$^2$) centered in the in situ sampling location, with time windows of ±3 h from 12:00 in the Strait of Georgia, ±5 h for the west coast of Vancouver Island, and ±12 h for Line P (Figure 1B). These time windows were defined to be short enough to reduce the effects of temporal variability on the in situ data, particularly in more dynamic regions, yet large enough to allow for the greatest possibility of a match [16]. To minimize the impact of low-quality match-ups, the area of the Fraser River plume was excluded since high turbidity has been shown to result in inaccurate chlorophyll estimates [17,18,20], and the high spatial variability of plume–ocean transitional waters can compromise the results of satellite validation [41]. For effective comparison, only match-ups with coincident valid pixels for all three satellite products were considered. To determine the relationship between the in situ Chl-a measurements and the product Chl-a values, a linear regression was performed on log-transformed data and the Pearson correlation coefficient (r-value) was calculated to determine the strength of the linear association between the two variables (Figure 2) [42]. To quantitatively evaluate the performances of the merged satellite products, metrics for the differences between the observed and measured values (root mean square error (RMSE) and median absolute difference (MdAD)), the mean bias (BIAS), and the goodness of fit (coefficient of determination (r$^2$) and regression slope) were calculated for log-transformed Chl-a data [43]:

$$\text{RMSE} = \sqrt{\frac{\sum_{i=1}^{n}(log_{10}(Sat_i) - log_{10}(InSitu_i))^2}{n}} \tag{1}$$

$$\text{MdAD} = 10^{Median|log_{10}(Sat_i) - log_{10}(InSitu_i)|} \tag{2}$$

$$\text{BIAS} = 10^{\frac{\sum_{i=1}^{n} log_{10}(Sat_i) - log_{10}(InSitu_i)}{n}} \tag{3}$$

where $Sat_i$ is the satellite-derived data, $InSitu_i$ is the in situ measurement, and $n$ is the number of samples. Note that chlorophyll was log-transformed prior to calculating error metrics [43].

An additional analysis was performed to evaluate the data quality of the large-scale dynamics of the study region. Seasonal trends were derived from the 23-year (1998–2021)

climatology of Chla-GCint data, which was separated into averages for spring (February–May; Figure 3A), summer (June–August; Figure 3B), and fall (September–November; Figure 3C) [18]. Latitudinal transects were extracted from the seasonal composites of chlorophyll concentration along the coastline (11 km from the coast) and over the outer margin of the continental shelf (following the 800 m isobath, which is ~65 km from the coastline; Figure 1A). The transect data were averaged from a 3 × 3 pixel window of the data, representing a 12 km wide band that was averaged along each latitude. The resulting latitudinal trends (Figure 4A–C) were verified according to known Chl-a seasonal and latitudinal dynamics over the study region.

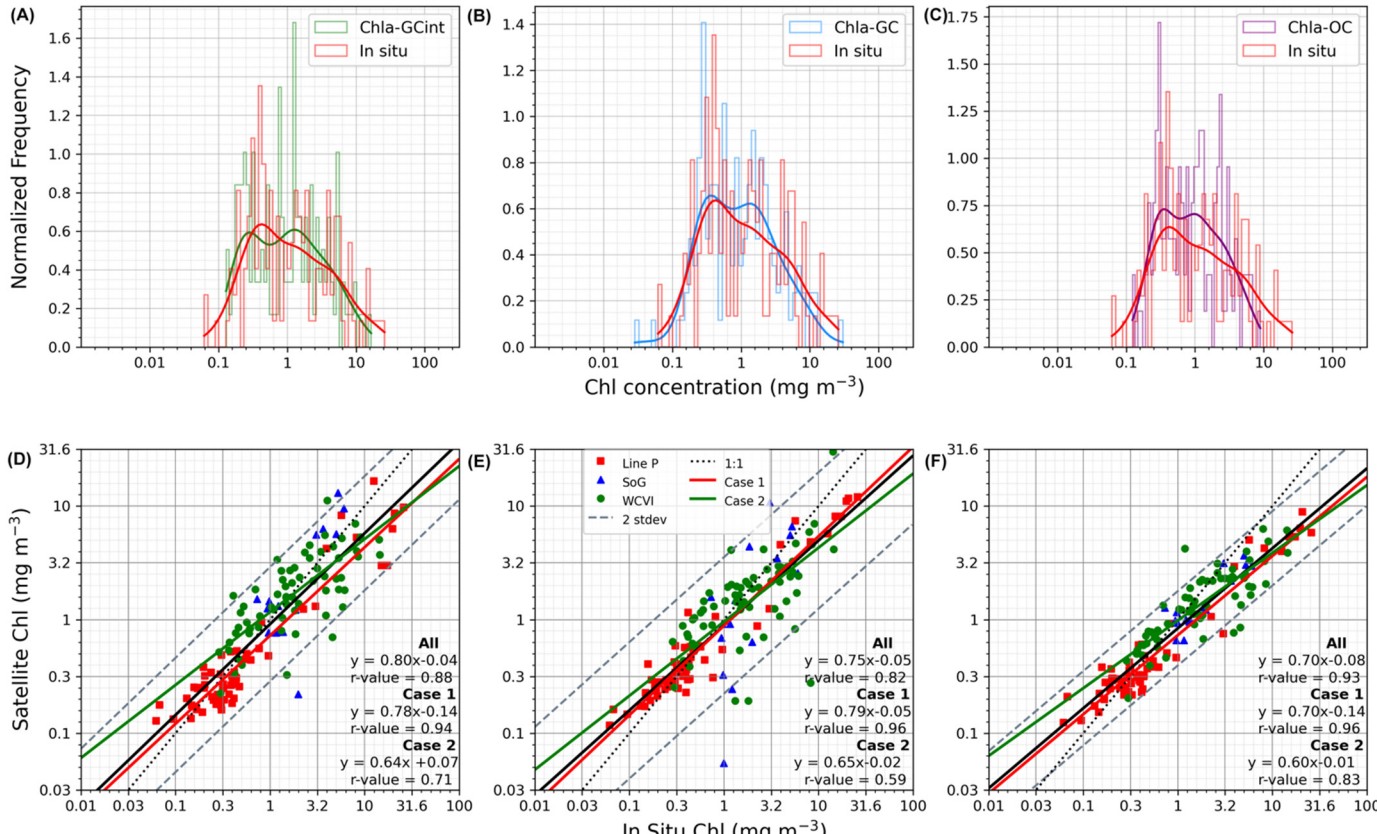

**Figure 2.** The top row shows the normalized frequency distributions of the Chl-a match-up measurements for (**A**) GlobColour interpolated (green), (**B**) GlobColour CHL1 (blue), and (**C**) OC-CCI (purple) with coincident in situ measurements (red). The lines show a stepped histogram with 50 bins, while the curves represent a kernel density estimate using Gaussian kernels. The bottom row shows the (**D**) GlobColour-interpolated-, (**E**) GlobColour-CHL1-, and (**F**) OC-CCI-derived Chl-a measurements for in situ Chl-a match-up scatterplot comparisons. The red, green, and blue markers represent data from the Line P, the west coast of Vancouver Island (WCVI), and the Strait of Georgia (SoG) locations, respectively. The black line represents the best fit line, the black dotted line is the 1:1 line, and the grey dotted lines represent 2 standard deviations of the best fit line. The green and red lines represent Case 1 waters (Line P) and Case 2 waters (SoG and WCVI), respectively. Data were log₁₀-transformed for display. Note that the top row (ABC) is in logarithmic scale, with log values converted into actual concentrations of Chl-a (mg m⁻³) on the x-axis labels, and the second row (DEF) are in logarithmic scale with log values converted into actual concentrations of Chl-a (mg m⁻³) on the x- and y-axes.

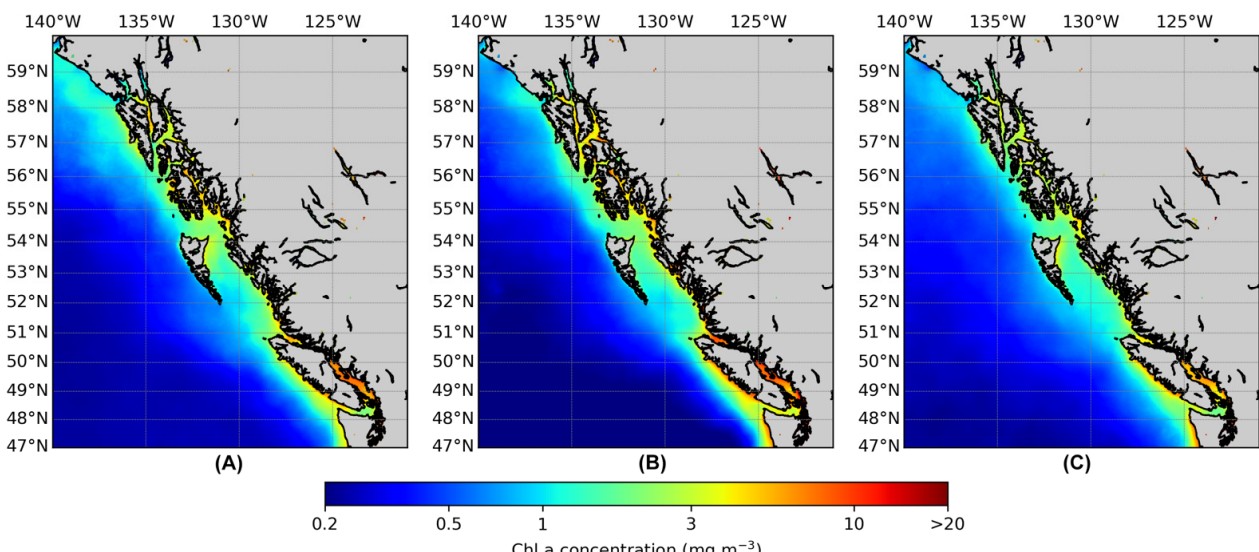

**Figure 3.** Seasonal climatology derived from GlobColour Interpolated Chl-a data for (**A**) spring (February–May), (**B**) summer (June–August), and (**C**) fall (September–November).

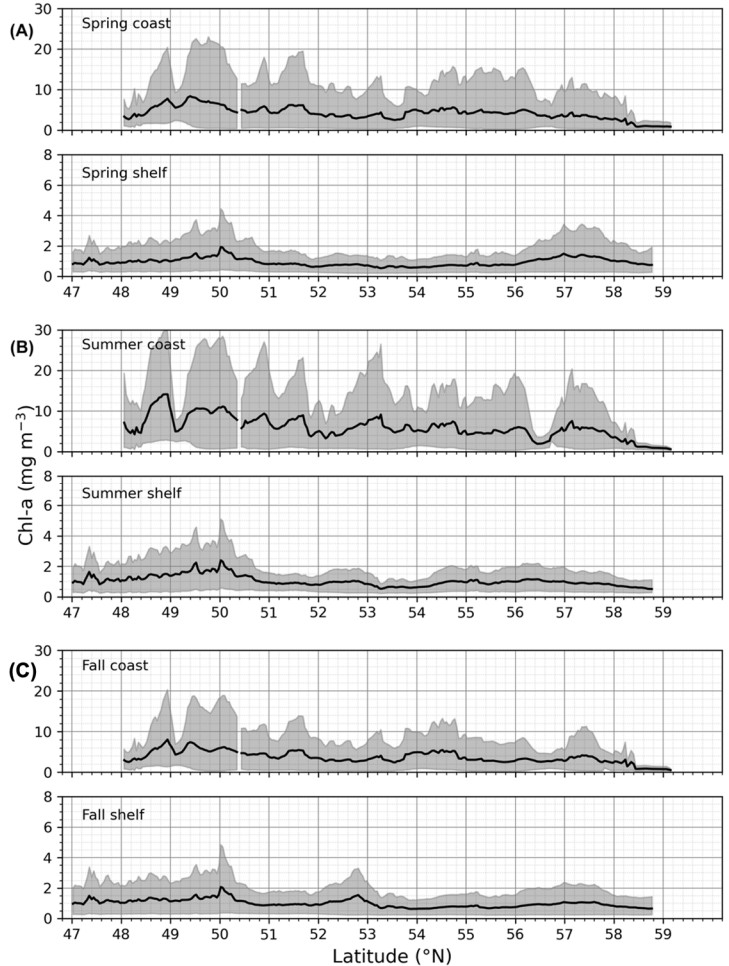

**Figure 4.** Chl-a latitudinal distribution of GlobColour Interpolated seasonal averages for (**A**) spring, (**B**) summer, and (**C**) fall along the coast and continental shelf transects in British Columbia and southeast Alaska. Gray shaded area represents the 10–90 percentile. Note the different y-axis scales between the coast and shelf plots.

## 3. Results

### 3.1. Statistical Analysis

The performance of Chla-GCint was evaluated by comparing satellite-derived Chl-a to in situ Chl-a concentrations via a one-to-one match-up analysis and, subsequently, comparing the statistics to the merged satellite products of Chla-GC and Chla-OC, as described in Section 2.3. The in situ Chl-a measurements for the match-ups ranged from 0.04 to 40.4 mg m$^{-3}$, reflecting the expected annual conditions experienced in British Columbia waters [29]. From the total 1914 in situ measurements, the common match-ups resulted in 141 for all the water types, with 59 for Case 1 waters (Line P match-ups; Figure 1B) and 82 for Case 2 waters (west coast of Vancouver Island and Strait of Georgia match-ups; Figure 1B). Figure 2A–F and Table 1 show the results for the common match-ups for each Chl-a product.

The Chla-GCint match-up analysis resulted in data that were closely concentrated around the best fit line with very few values exceeding two standard deviations (RMSE = 0.28, r = 0.88; Figure 2E), a slope relatively close to unity (slope = 0.80), a median percentage difference of 50% (MdAD = 1.5), and a model underestimation of 10% (BIAS = 0.90; Table 1). Additionally, 77% of the in situ Chl-a variability was predictable by the Chla-GCint observations (r$^2$ = 0.77). The distribution of the Chla-GCint observations indicated a slight underestimation compared with the in situ values (Figure 2A). The regression line crossed the 1:1 line at −0.2 log$_{10}$, meaning that Chla-GCint underestimated (overestimated) values greater (less) than 0.6 mg m$^{-3}$ (Figure 2E). Further, the Chla-GCint data points showed a clear pattern based on the sampling region. The data points for Case 1 waters were very concentrated at lower Chl-a concentration values compared to the data points for Case 2 waters, which yielded higher concentrations and a greater spread, meaning that the match-ups in coastal waters were more variable (Figure 2D).

Considering the noninterpolated products, overall, Chla-OC was the best-performing product in most statistics (r = 0.93, r$^2$ = 0.81, RMSE = 0.25), apart from slope, BIAS, and MdAD. Chla-OC and Chla-GC shared the same median percentage difference of 43% (MdAD = 1.43). Overall, the statistics of Chla-GCint were only marginally inferior to the best-performing product for all the water types (Chla-OC), and in the cases of slope, intercept, and BIAS, Chla-GCint outperformed Chla-OC and Chla-GC.

When considering the product performance in Case 1 waters, the statistics of all the products were generally improved compared to the statistics obtained for all the water types (Table 1). The best-performing product in Case 1 waters was Chla-GC; Chla-GCint had a similar r-value and slope and inferior but comparable values for BIAS, MdAD, and RMSE (Table 1). For Case 2 waters, the best-performing product was Chla-OC for most of the statistics (r = 0.83, r$^2$ = 0.62, MdAD = 1.44, RMSE = 0.26; Table 1), although Chla-GCint was only marginally inferior to Chla-OC, and in the cases of slope and BIAS, Chla-GCint outperformed Chla-OC. Additionally, Chla-GCint outperformed Chla-GC in all the statistics for Case 2 waters (Table 1).

Overall, when comparing Chla-GCint to the best-performing product for each water type, the statistics were only marginally inferior, and the statistical metrics were comparable. Identical to Chla-GCint, Chla-GC and Chla-OC underestimated (overestimated) values greater (less) than 0.6 mg m$^{-3}$ (Figure 2E,F); however, Chla-OC overestimated (underestimated) at a slightly greater degree, with a slope of 0.70 (Figure 2F). The distribution of Chla-GC followed the in situ Chl-a data distribution very closely up to about 0.6 mg m$^{-3}$ and slightly overestimated the higher range of Chl-a; it began to diverge thereafter (Figure 2B). Similarly, the Chla-OC distribution followed the in situ curve, although not as closely as Chla-GC, and diverged at a lower Chl-a value of 0.3 mg m$^{-3}$ (Figure 2C).

**Table 1.** Statistical output comparing product performance for OC-CCI, GlobColour CHL1, and GlobColour interpolated match-up analysis. The shaded cells indicate which product performed best for each water type.

| | All | | | Case 1 | | | Case 2 | | |
|---|---|---|---|---|---|---|---|---|---|
| | Chla-GCint | Chla-GC | Chla-OC | Chla-GCint | Chla-GC | Chla-OC | Chla-GCint | Chla-GC | Chla-OC |
| n | 141 | 141 | 141 | 59 | 59 | 59 | 82 | 82 | 82 |
| r | 0.88 | 0.82 | 0.93 | 0.94 | 0.96 | 0.96 | 0.71 | 0.59 | 0.83 |
| $r^2$ | 0.77 | 0.66 | 0.81 | 0.86 | 0.91 | 0.86 | 0.47 | 0.02 | 0.62 |
| slope | 0.80 | 0.75 | 0.7 | 0.78 | 0.79 | 0.70 | 0.64 | 0.65 | 0.6 |
| intercept | −0.04 | −0.05 | −0.08 | −0.14 | −0.05 | −0.14 | 0.07 | −0.02 | −0.01 |
| *p*-value | <0.001 | <0.001 | <0.001 | <0.001 | <0.001 | <0.001 | <0.001 | <0.001 | <0.001 |
| BIAS | 0.90 | 0.87 | 0.82 | 0.83 | 1.006 | 0.88 | 0.96 | 0.79 | 0.78 |
| MdAD | 1.5 | 1.43 | 1.43 | 1.44 | 1.32 | 1.38 | 1.57 | 1.65 | 1.44 |
| RMSE | 0.28 | 0.34 | 0.25 | 0.24 | 0.20 | 0.24 | 0.30 | 0.41 | 0.26 |

*3.2. Seasonal and Latitudinal Trends*

The Chla-GCint product underwent further analysis to evaluate the quality of this product to capture known spatial and temporal patterns of bloom dynamics for this region. Seasonal trends corresponding to spring, summer, and fall were derived from the 23-year (1998–2021) Chla-GCint data series. The retrieved temporal trends showed that, on the B.C. coast, the maximum bloom concentrations were observed in the Strait of Georgia in spring and summer, and relatively high concentrations were experienced on the west coast of Vancouver Island and Queen Charlotte Strait in summer (Figure 3A,B). In fall, a secondary phytoplankton bloom was observed, with the highest concentrations occurring in the Strait of Georgia, on the west coast of Vancouver Island, and Washington coast (Figure 3C). The central and northern B.C. coastal waters had lower concentrations on average, with the lowest surface chlorophyll concentrations observed off the west coast of Haida Gwaii (Figure 3A–C).

Spatially, transects of Chl-a seasonal averages along the continental shelf and coastline revealed that, in the B.C. coastal region, Chl-a values ranged from 0.51 ± 0.6 to 14.1 ± 11.2 mg m$^{-3}$ (Figure 4A–C). The maximum coastal Chl-a values were found in the Strait of Georgia (~48.5°N–50°N) in spring (8.4 ± 10 mg m$^{-3}$ at ~49.4°N; Figure 4A, coast), summer (14.1 ± 11.2 mg m$^{-3}$ at ~48.9°N; Figure 4B, coast), and fall (8.1 ± 6.8 mg m$^{-3}$ at ~48.94°N; Figure 4C, coast). In contrast, the lowest Chl-a coastal values were found along Johnstone Strait (50.2–50.5°N, Figure 4C, coast), as well as generally low Chl-a values in the northern southeast Alaska waters (north of 58.4°N, Figure 4A–C, coast).

**4. Discussion**

This study provides an evaluation of the interpolated Chl-a product from GlobColour (Chla-GCint), and compares the statistical validation to merged multi-sensor Chl-a products from OC-CCI (Chla-OC) and GlobColour CHL1 (Chla-GC) in the coastal and open ocean waters off southern B.C. This is an important step prior to using this product for monitoring applications such as phytoplankton phenology and bioregionalization studies, since regional optical complexity makes the application of ocean colour remote sensing challenging and can introduce accuracy biases [7,16–18,22].

*4.1. Statistical Validation*

The comparison of satellite-derived Chl-a with in situ measurements suggests that the Chla-GCint product performed relatively well in this region and was comparable to the Chla-OC and Chla-GC products. The Chla-OC product had the best overall statistics for all the water types and Case 2 waters, whereas Chla-GC had the best statistics in Case 1 waters. It is generally expected for interpolated products to have degraded statistics due to the estimations of interpolated values being based on assumption [19,44], causing Chl-a values to diverge from in situ Chl-a. Despite having different absolute Chl-a

concentrations ($r^2 = 0.77$, Table 1), the Chla-GCint product shared a similar distribution as the in situ data (Figure 2A) and had fewer missing data than any other global time series, which makes it suitable for use in studies concerning phytoplankton phenology where Chl-a trends and seasonality are prioritized over absolute concentrations.

### 4.1.1. Merging and Flagging Strategies

Generally, the differences between Chl-a retrieval methods for different products can be attributed to the characteristics of the products themselves, such as the choice and performance of the atmospheric correction scheme, the ocean colour inversion algorithm and merging techniques applied, or the spatial and temporal quality of the match-ups, including uncertainties associated with in situ Chl-a data, as well as the chosen statistics. Prior to interpolation, the GlobColour interpolated product was produced by first merging Chl-a fields estimated using different algorithms (OC3 and OC5 for Case 1 and 2 waters, respectively) generated from Level 2 reflectance [38]. Since OC-CCI and GlobColour use the same Chl-a algorithms (CI and OC5), the differences in their performance could be attributed to atmospheric correction, merging strategies, and flagging schemes [8]. Atmospheric correction is employed to derive remote-sensing reflectance of the sea surface from the top-of-atmosphere radiance. OC-CCI uses the POLYMER atmospheric correction method for all sensor data besides SeaWiFS, where NASA's Level 2 Generator processor is applied [45]. The merging strategy used by OC-CCI is based on the preliminary merging of the remote-sensing reflectance of a set of sensors, which is then used to derive Chl-a. OC-CCI derives Chl-a using a blended Chl-a algorithm that attempts to weight the outputs of the best-performing algorithms based on the water types present [8]. Alternatively, GlobColour first computes Chl-a for each sensor using their specific resolutions and spectral bands and, subsequently, resamples and merges the single-sensor Chl-a products [8]. However, each sensor has its own atmospheric correction procedure, with varying levels of success [46]. Further, the continuities for GlobColour algorithms used for mesotrophic and complex waters are determined by the OC5 lookup table, and when Chl-a concentrations range from 0.15 to 0.2 mg m$^{-3}$, a linear interpolation of OC5 and CI is used [8].

Beyond the different approaches to generate a merged Chl-a product, different flagging strategies are used. GlobColour CHL1 and the interpolated product [10] use the OC5 flagging strategy, which employs an algorithm that uses the official flags and empirical thresholds adjusted for each sensor. OC-CCI uses a more constrained flagging strategy that depends on the sensor—for instance, a pixel classification algorithm (Idepix) is implemented for MERIS and SeaWiFs data processing in v2, with NASA's Level 2 Generator being used for the other sensors [47].

### 4.1.2. Spatial and Temporal Dynamics

Beyond the different atmospheric corrections, merging strategies, and flagging schemes applied to these products, differences in the spatiotemporal scale of sampling between the satellite and in situ measurements need to be considered in the analysis of the Chl-a accuracy of each product. The GlobColour and OC-CCI product footprints (16 km$^2$) cover an area that is orders of magnitude larger than that captured by the in situ measurements (<1 m) [48,49]. The spatial variability of coastal waters also affects the quality of the in situ data to be used in a match up, where data acquired from interface or transition waters are generally of a lower quality [41,50].

Additionally, the temporal aspect of the match-up time difference between in situ data and satellite overpasses poses considerable uncertainties for the validation of satellite-derived products, especially for coastal waters. Coastal waters are highly dynamic and have greater optical complexity due to the influences of river discharge containing terrestrial suspended particulates, resuspended sediments, and CDOM that vary independently of the phytoplankton assemblage [21–23,41,51,52]. To account for this, the area of the Fraser River plume was excluded since it is very temporally dynamic due

to tide, current, and river discharge conditions, which can result in inaccurate chlorophyll estimates [17,18,20,41].

GlobColour and OC-CCI products do not contain a timestamp; however, products are obtained from sun-synchronous satellites having crossing nodal times at the equator between 9:50 to 13:30 (PDT) depending on the satellite, which is why an approximate time of 12:00 was used in this analysis and for GlobColour global validation [14]. Temporal match-up windows of ±3, ±5, and ±12 h intervals around midday were applied to the Strait of Georgia (estuarine), the west coast of Vancouver Island (coastal), and Line P (open ocean; Figure 1B), respectively, in order to reduce the effects of temporal variability on the in situ data, particularly in more dynamic regions [16]. The longer time window of 12 h is acceptable for open ocean waters since the subarctic north Pacific is a high-nitrate low-chlorophyll region (HNLC) controlled by large-scale currents with minimal seasonal variability of primary productivity and is relatively more homogenous than coastal waters [53,54]. While these specific conditions were taken into consideration to minimize the match-up times in more dynamic regions (Section 2.3, inconsistencies could still occur because in situ collection and satellite overpass did not occur simultaneously.

### 4.1.3. In Situ Measurements

Other sources of uncertainty that could impact the validation results include errors associated with the analytical quality of the in situ measurements. Despite being used as 'ground truth' measurements, they are seldom 'absolute truth', and their uncertainties should be recognized [16]. The quality of in situ data is dependent on measurement protocols involving sampling, filtration, storage, extraction, HPLC analysis, instrument calibration, and deployment, to name a few [48]. In an attempt to reduce systematic uncertainties pertaining to data processing, it is advised that data are consistently processed using a single-source processor [16], which is why HPLC data processed solely by DFO was used here.

### 4.1.4. Statistical Metrics and Global or Regional Validation

Lastly, the metrics used for the statistical analysis need to be carefully chosen to suit the data type. BIAS and MdAD are metrics based on simple deviations that are generally well-suited for evaluating non-Gaussian distributions and outliers and, therefore, take precedence over the interpretation of RMSE, the coefficient of determination ($r^2$), and regression slopes, which are most appropriate for Gaussian distributions with outliers, making them suboptimal metrics to determine ocean colour algorithm assessment [43]. Additionally, slope and $r^2$ are useful metrics for ocean colour validation; however, they must be interpreted in the context of BIAS and MdAD, as these metrics can be misleading when interpreted in isolation [43]. Here, we took these precautions into consideration by giving precedence to BIAS and MdAD and interpreting the other metrics in the context of them.

Given the many sources of uncertainties associated with these merged data products, many studies (Table 2) have performed regional comparisons to in situ Chl-a concentrations to ensure that the products can be applied to different waters, despite these merged multi-sensor products having been already validated on a global scale. Table 2 provides a summary of some studies that have undertaken regional validation of merged GlobColour–OC-CCI Chl-a data in different regions, study periods, and water types to compare the validation statistics obtained here. For instance, the analysis undertaken by Swart et al. (2012) over the Good Hope line south of Africa showed a similar $r^2$ value of 0.84 (Table 2) but a slightly higher RMSE value of 0.50 [55]. Pitarch et al. (2016) also performed a comparison between GlobColour and OC-CCI in the Baltic Sea, where OC-CCI OC5 emerged as the best-performing product and GlobColour performed the worst in that region [56]. Nonetheless, the OC-CCI OC5 statistics defined by Pitarch et al. (2016) are comparable to the OC-CCI Case 2 statistics defined here [56]. Validations of Case 1 waters generally have better statistics than those derived in Case 2 waters (Table 2). Many

studies have not performed regional validations prior to utilizing data products in their research [7,8,12,57–59], and of other studies that have performed validations, the statistics obtained are generally inferior to the validation statistics derived here (Table 2). Compared to regional studies that utilize different satellites and implement different methodologies, the statistics obtained here are comparable to, and in some cases outperform, the statistics obtained in other local studies. For instance, the GlobColour statistics obtained here generated lower BIAS and RMSE values and a larger r-value than Carswell et al. (2017), as well as greater r-value and slope values and lower BIAS and MdAD values than Giannini et al. (2021), attesting to the quality of the GlobColour-derived product [17,18]. Additionally, when comparing Case 2 GlobColour interpolated validation statistics with those derived from a data-interpolating empirical orthogonal function (DINEOF) 'gap filled' dataset in the Salish Sea [19], the $r^2$, RMSE, and slope results of Chla-GCint outcompeted the statistics found by Hilborn and Costa (2018) (Table 2) [19]. However, different interpolating methods were used to derive the products (DINEOF vs. standard optimal interpolation technique) and, because Hilborn and Costa (2018) did not use merged products, the data gap was larger than the input data used for the GlobColour interpolated product [19].

Considering the match-up statistics for Case 1 and Case 2 waters individually, our results indicated how the quality of products can vary in different environmental conditions, highlighting the need for the continuous assessment of satellite-derived Chl-a products, particularly in optically complex waters [7,16,18–20,55]. The results for Case 1 waters showed improved statistics for Chla-GCint, Chla-GC, and Chla-OC since the Chl-a algorithms were modeled for these water types [6,47]. The statistics obtained for Case 2 waters for OC-CCI (Table 1) were similar to those obtained by the OC-CCI OC5 algorithm from Pitarch et al., 2016 [56]. The statistics for all the products were degraded in Case 2 waters, thus highlighting the challenges of ocean colour remote sensing and, particularly, the application of algorithms designed for Case 1 waters in these Case 2 waters.

**Table 2.** Nonexhaustive list of research using merged GlobColour–OC-CCI Chl-a data in different regions, study periods, and water types. Table is separated into research that validated the Chl-a data prior to use and the reported statistics, followed by those that did not validate the products with in situ Chl-a data. The column of 'Similar Method' refers to whether the data were log-transformed and whether regression parameters or statistical metrics were calculated in a similar way as here, making the results comparable.

| Author | Product | Time | Location | Case 1 or 2 | N | R² | RMSE | BIAS | Similar Method |
|---|---|---|---|---|---|---|---|---|---|
| | | | | Validated | | | | | |
| [55] Swart et al., 2012 | GC-OC4 | 2010–2011 | Good Hope line south of Africa | 1 | 121 | 0.84 | 0.50 | | Yes |
| [60] Laiolo et al., 2021 | GC-GSM | 2016 | offshore eastern Australia ocean region | 1 | 9 | | | 0.63 | Yes |
| [61] Johnson et al., 2013 | GC-AVW | 2001–2008 | southern Ocean | 1 | | 0.25 | | | No |
| [62] Gbagir & Colpaert, 2020 | GC CHL-OC5 | 1997–2019 | Lake Ladoga | 2 | | Not reported | | | N/A |
| [56] Pitarch et al., 2016 | GC-GSM | 1997–2012 | Baltic Sea | 2 | 1873 | 0.3 | 0.42 | 0.77 | Yes |
| | OC-CCI OC5 | | | | 1873 | 0.44 | 0.28 | 0.86 | Yes |
| | OC-CCI OC4v6 | | | | 1873 | 0.43 | 0.33 | 1.44 | Yes |
| [37] Moradi, 2021 | GC-CHL2 | 2008–2018 | Persian Gulf | 2 | 275 | 0.41 | 0.53 | 4.15 | Yes |
| | OC-CCI | | | | 487 | 0.44 | 0.49 | 0.38 | |
| [63] Cherkasheva et al. (2014) | GC-GSM | 1998–2009 | Fram Strait and Greenland Sea | 1 and 2 | 108 | 0.34 | 0.58 | | No |
| [64] El Hourany et al., 2019 | GC CHL-OC5 | 1997–2014 | Global | 1 and 2 | | 0.49 | | | No |

| [19] Hilborn & Costa 2018 | DINEOF interpolated | 2014–2016 | Salish sea | 2 | 45 | 0.23 | 0.39 | Yes |
|---|---|---|---|---|---|---|---|---|

### *4.2. Seasonal and Latitudinal Trends*

Despite the statistics obtained here, considering all the possible challenges mentioned hitherto and the caution taken to minimize these issues, it remains complicated to truly evaluate the uncertainties associated with global products. As such, a general comparison with the Chl-a trends published in the literature also provides a tool for product evaluations. Therefore, to further evaluate the application of the GlobColour interpolated Chl-a product for British Columbia and southeast Alaska, the seasonal and latitudinal ranges of variability in Chl-a were compared with those previously reported for different parts of the coast (Figure 4A–C), particularly since the considered in situ match-up dataset was mostly constrained to the southern B.C. coast due to data availability. The GlobColour interpolated Chl-a product demonstrated the expected seasonal and local dynamics for this region, as well as average concentrations within ranges reported for satellite-derived observations [17,18,29,30,65,66] and in situ measurements [32]. For instance, this region typically experiences maximum Chl-a concentrations in the spring and summer [17,18,29,30,32,67,68], with a secondary fall bloom being observed in the B.C. coastal waters [18,30,69], which was particularly evident in the vicinity of Calvert Island (51.5°N; Figure 4C, coast). Along the continental shelf, the west coast of Vancouver Island experienced relatively higher Chl-a concentrations (maximum of 3.4 ± 3.9 mg m$^{-3}$; 50°N, Figure 4A–C, shelf), consistent with reported concentration ranges and bloom timing [18,69]. The B.C. central shelf region between Haida Gwaii and Vancouver Island showed lower Chl-a concentrations overall compared to the southeast Alaska shelf region, which is typically shown to be relatively more productive in summer [29].

### 5. Conclusions

The performance of the GlobColour interpolated Chl-a product was evaluated in B.C. waters using in situ data to determine whether it was suitable for use in this region, followed by comparing the statistical results to OC-CCI and GlobColour merged Chl-a products. An additional qualitative analysis was conducted to determine whether the data reflected the region's large-scale seasonal trends and latitudinal dynamics. The satellite and in situ match-up analysis in this region revealed that the GlobColour interpolated product performed relatively well and was comparable to the best-performing product for each water type, with OC-CCI as the best performer across most metrics in all the water types and Case 2 waters and GlobColour CHL1 as the best performer in Case 1 waters. The GlobColour interpolated dataset also reflected the expected general seasonal and latitudinal averages over the entire study region. Compared to other studies, the GlobColour interpolated statistics obtained here were comparable to, and in some cases outperformed, other validation statistics. The GlobColour interpolated product had the highest spatial coverage since it was 'gap-filled' and was concluded as appropriate for use in studies requiring a spatially and temporally complete dataset with a slight disregard for absolute Chl-a concentration, such as phytoplankton phenology studies. Although several studies have shown that the OC-CCI and GlobColour Chl-a datasets can be implemented in coastal turbid waters to monitor Chl-a concentrations [37,48,61,70], the application of these products in primarily Case 2 waters needs to be performed with caution, considering the reduction in statistical performance shown in this study. It is, therefore, recommended that a comparison between Chl-a products and in situ water samples should generally be implemented to aid product selection before further integration into analysis.

**Author Contributions:** Conceptualization, S.P., J.M.J. and M.C.; methodology, S.P., J.M.J., M.C. and M.K.; formal analysis, S.P.; writing—original draft preparation, S.P.; review and editing, M.C., J.M.J. and M.K.; supervision, M.C. and J.M.J., funding acquisition, M.C. All authors have read and agreed to the published version of the manuscript.

**Funding:** This research was supported by Costa with funds from Canadian Space Agency (FAST 18FAVICB09), the NSERC NCE MEOPAR—the Marine Environmental Observation, Prediction, and Response Network, and the NSERC Discovery Grant, Canada.

**Data Availability Statement:** The GlobColour datasets analysed in this study can be found at http://hermes.acri.fr. The interpolated GlobColour dataset analysed in this study can be found at https://resources.marine.copernicus.eu/product-detail/OCEANCOLOUR_GLO_CHL_L4_REP_OBSERVATIONS_009_082/DATA-ACCESS. The OC-CCI dataset, Version [v5.0], ESA, analysed in this study can be found at http://www.esa-oceancolour-cci.org/.

**Acknowledgments:** The GlobColour data (CHL1, GSM; http://globcolour.info (accessed on 12 April 2021) used in this study were developed, validated, and distributed by ACRI-ST, France, and are available at Hermes (http://hermes.acri.fr). Further details on the GlobColor dataset can be found at https://www.globcolour.info/CDR_Docs/GlobCOLOUR_PUG.pdf. This work is a contribution to the NSERC DG to Costa; MEOPAR—Marine Environmental Observation, Predication, and Response Network; and the Canadian Space Agency (FAST 18FAVICB09) project's "Use of ocean colour satellites to characterise waters along the migration route of juvenile salmon in British Columbia and Southeast Alaska" to Costa.

**Conflicts of Interest:** The authors declare no conflict of interest.

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
