# Peer review of "Merged Multi-Sensor Ocean Colour Chlorophyll Product Evaluation for the British Columbia Coast"

_remotesensing, doi:10.3390/rs15030687_

Round 1

Reviewer 1 Report (Previous Reviewer 2)

The authors have made several changes to the manuscript based on my major comments. The revised version of the manuscript has been improved significantly both in terms of results and presentation.

I still have a few minor comments on the revised manuscript,

Line 44: A space is needed before “[4]”.

Fig. 1B: It has been stated in the responses and the manuscript that some stations from highly dynamic regions of the Strait of Georgia and the Fraser River have been removed from the analysis. However, I think, they haven’t been removed from fig. 2b. Please correct this if this is true and only show 141 stations that are used in the analysis.

Line 116: There is an extra space between two words.

Line 154: A space is needed before “[6]”.

Lines 186-189: They can be removed.

Line 207: “accuracy”. Both RMSE or RMSD (root mean squared difference) and MdAD are the metrics for the difference between the observed and measured values. Accuracy of a measurement refers to the closeness of agreement between a measured quantity value and the (unknown) true value of the measurand (IOCCG 2019, VIM 2012).

Eq.3: Please remove “^” from the equation.

Fig. 2: Half of figures 2c and 2f are missing for the PDF version.

Line 367: I would encourage authors to add a few lines on how the atmospheric correction schemes are different among various merged products.

Line 427: This is partially true. Because BIAS is the mean bias in this study and the outliers or non Gaussian distribution of differences can significantly affect the average statistics.

Table 2: What is the point of including these studies in the table when they do not provide any statistics to compare to the current study? I would move remove them from the table and cite them somewhere near lines 449-451.

Line 524: Extra space.

Author Response

Reviewer 2 Report (New Reviewer)

The manuscript describes a validation exercise relying on GlobColour interpolated Chla concentrations as provided by Copernicus Marine, and focusing on the coast of British Columbia. GlobColour daily observations (CHL1) and OC-CCI Chla have been also included for benchmarking.

Satellite measurements are firstly compared to a set of in situ measurements by applying widely used statistical indicators, and then qualitatively evaluated in terms of their seasonal variability with respect to the known regional biogeochemical dynamics.

Results indicate similar performances for all the satellite products - i.e., medium absolute misfits but similar trends wrt in situ data - with a slight preference towards GlobColour interpolated for being “gap-filled” and thus more useful in phenological studies.

The manuscript is generally clear, except for the Results section, where the comparison between the results obtained with the different products is very hard to follow.

As a general comment, I'd suggest that the authors carefully review the naming convention through all the manuscript, e.g, GC vs Chla-GC vs GlobColour CHL1; OC vs Chla-OC vs OC-CCI; GCint vs Chla-GCint vs GlobColour interpolated; Chl vs Chla; r (table) vs r-value(text and figure).

Analogously, to be consistent with the description of the results, please update Table 1 so that GCint (of Chla-GCint) is at the first column for each water type.  

Detailed Comments:

Results section:
-    how are Case-1 and Case-2 waters distinguished?
-    The concept of coefficient of determination should be applied in the opposite direction. In situ measurements are the independent variable, while the satellite measurements are the dependent one. So 77% of the in situ variability is caught (or predictable) by the satellite observations.  
-    MdAD at line 256 should be 1.43. Also, I would not use the terms “error” but rather say that MdAD is an indicator of the dispersion of the differences sat-in situ, so higher values of MdAD indicate higher dispersion.   
-    Sentence at lines 256-257 is repeated
-    As mentioned, all the paragraph between lines 260 and 275 is very hard to follow.
-    Figure 2 is not fitting the page. Use “Satellite Chla” instead of “Product Chla” on the y axis. Please redo Figs 2d 2e and 2f in logscale, as 2a, 2b, and 2c.

Discussion section:
Except for the first paragraph which summarizes obtained results, the rest of the paragraphs do not add much to the paper, most of the comments are very generic and no extra comparison or specific test is included to support those statements. For example: comments about the merging techniques (lines 362-375) were already given in section 2 as well as those about the choice of the insitu measurement dataset; the impact of spatial variability and the temporal difference on satellite matchups are well known and documented, a good extension would be to include an additional test using the same validation dataset but changing temporal and/or spatial windows.

Author Response

This manuscript is a resubmission of an earlier submission. The following is a list of the peer review reports and author responses from that submission.

Round 1

Reviewer 1 Report

The OC-CCI product is a useful longterm multi-sensor merged products readily for applications. The authors carried out a compresensively regional evaluation of the Chlorophyll product, which could provide a reference for product selection and thus faciliate their applications. Overall, the topic is interesting and the paper is well-written. However, there are some issues to be revised before it can be considered for publication. Therefore, I suggest a major revision.
1. Suggest to reorganize the Introduction section. The three paragraphs are too long for readers to well follow the logics.

2. Also suggest to reorganize other long paragraphs.

3.  The authors put all the figures and tables in a single section 3.3. This is rarely be seen in other publications. Suggest to arrange them where you mention them in the context.

Reviewer 2 Report

The manuscript entitled “Merged multi-sensor ocean colour chlorophyll product evaluation for the British Columbia coast” by Pramlall et al. presents the intercomparison analysis to assess the performance of three merged (level 3) daily chlorophyll-a products, GlobColor CHL1, GlobColour interpolated, and OC-CCI using a suite of in situ chlorophyll-a measurements that are collected from coastal and open ocean waters of British Columbia between 2006 and 2017. The intercomparison analysis shows that the GlobColor CHL1 product provided relatively better estimates of Chlorophyll-a concentration in the study area in comparison to the other two chlorophyll products whereas the GlobColor interpolated chl-a product performed worst among three candidates despite having the highest number of matchups available for the assessment for this product. When the products are evaluated for different water types, a general agreement was observed among them with their performances remaining relatively better in Case 1 water than the Case 2 water. The best performing chl product, GlobColour CHL1 is also used to examine seasonal and latitudinal trends in chlorophyll-a in the study area.

             I have major concerns about the approaches that have been used to obtain the matchup corresponding to the field station and to determine the superiority of the chlorophyll product in this study (See my major comments).

Major comments:

Line 210 – 213: Why has a 3 x 3-pixel window centered at the in situ sampling station been used for this analysis? The chlorophyll values of 16 native pixels in the L2 scene (e.g., each with 1 km spatial resolution for MODIS-Aqua) are already averaged to obtain L3 product with a pixel size of 4 km x 4 km. The comparison of in situ chlorophyll concentration seems more reasonable with the chlorophyll concentration averaged over 16 km2 (4 km x 4 km pixel); however, it is a bit difficult to understand how the chlorophyll concentration averaged over an area of 144 km2 ( 12 km x 12 km) would represent the chlorophyll concentration that is observed at the in situ sampling location, especially for the shelf stations (e.g., WCVI and SoG in Fig. 1). Perhaps such averaging over 3 x 3-pixel box is valid for the off-shelf stations of Line P.

Section 3.1: For the assessment of individual global satellite datasets, it is OK to have a different number of in situ-satellite matchups corresponding to different satellite datasets (e.g., 124 for Chla-GC, 797 for Chla-GCint, and 201 for Chl-OC). However, it is important to remember that the aggregate statistical metrics described in section 2.3 represent a combined effect from all points of the individual dataset. More importantly, if outliers (large differences between satellite and in situ chlorophyll) are present in the given dataset, they would affect the mean statistics (i.e., RMSE, BIAS) and regression results of the dataset more heavily. The consistency in the comparison of the performance assessment of different datasets will be lost if such outliers are not present in the other two datasets.

Thus, for the intercomparison of three satellite chl products, it is required to have their assessment on the same number of stations (satellite-in situ matchups) to get a clear idea of the superiority of one dataset over the others. For example, Chla-GCint has the highest number of data to assess the performance of GCint chlorophyll product over the study area. Especially, figure 4a suggests that this product provides more match-ups in nearshore waters (e.g., blue and green symbol – Strait of Georgia and WCVI). However, several of these points are scattered around the 1:1 line. Moreover, the GCint product generally provides overestimated value of chlorophyll in the Strait of Georgia (blue triangles in Fig. 4a). These points would have a notable effect on aggregate statistics and regression results. Overall, such stations would degrade the aggregate statistics and regression results which can be seen in Figure 4 and Table 1 (i.e., relatively poorer BIAS, RMSE, and MdAD metrics). On the other hand, these stations are not present in the other two datasets and there is no way to determine how these datasets perform at the same statins in the waters of the Strait of Georgia or the near-shore region of WCVI. Thus, the inference of superior performance of one dataset over the others based on the results that are presented in Figure 4 and Table 1 would not do justice to the datasets having a relatively larger number of matchups than the GC dataset (i.e., the GCint and OC products).

Minor comments:

Line 40: Extra space between “from” and “different”.

Line 40: Extra space between “complex” and “because”.

Line 76: “accuracy biases”. Accuracy and bias have different meanings in statistics. I would suggest using different words to convey the message.

Line 86: Extra space between “27” and “300”.

Line 88: Extra space between “many” and “riverine”.

Line 117: The unit should be “mg m-3” instead of “mg.m-3”.

Line 123: Why only silicate? Are other nutrients such as nitrate and phosphate not responsible for limiting the phytoplankton growth in the region? Perhaps this sentence needs rewriting.

Line 151: “daily global data” itself provides information on the temporal resolution of the data products. Thus, the use of “daily temporal resolution” seems a repetition in line 152.

Line 159: It is VIIRS-SNPP (Suomi National Polar-orbiting Partnership).

Line 198: “VIIRS” instead of “VIIRs”.

Lines 215 – 218: If each pixel of the L3 product is matched to the chlorophyll concentration at the in situ sampling location, this step is not required.

Line 228; Eq. 2: The equation suggests that RMSE is computed using absolute values of in situ and satellite-derived chlorophyll concentrations. In this case, RMSE should have the unit of mg m-3. However, no unit has been used with the RMSE in a manuscript. Line 234 mentions that the chlorophyll was log-transformed prior to calculating error metrics. If true for RMSE, then it would be unitless. However, the equation needs to be corrected. If not true for RMSE, then units should be provided at various sections of the manuscript where the RMSE is mentioned.

Line 225: In this study, “bias” is the difference between satellite-derived Chl and measured Chl at a given station. However, Equation 4 suggests that it computes the “mean” bias using log-transformed values of satellite-derived chlorophyll and in situ chlorophyll concentrations. Thus, I would suggest using “Mean bias” instead of “BIAS” for this statistical metric.

Section 3.3: This section seems out of place. I would suggest keeping figures and tables close to the location where they are being discussed. Truly speaking, I have never come across any paper that has a separate section containing all figures and tables.

             Figure 1A: “Shelf (800 m)”. I would suggest using “Shelf (depth = 800 m)” or “Shelf (800 m isobath)” and “Coast (11 km from the coast)”.

Figure 1B: I would suggest using the full name of “SoG” and “WCVI” in the legend or the figure caption. Also, there are no lines in Figure 1B but the legends have both lines and points.

Table 1: The statistics in this table would change if intercomparison is done using a correct approach as described in my comments for Section 3.1. However, it can be kept in the manuscript to demonstrate the performance of an individual product instead of comparing different products.

Figure 2: I would suggest adding two transects (coast and shelf) to these figures because that has been discussed in detail in Figure 3 and section 3.2. “mg m-3” has been used in the text, thus please use the same in the figure to be consistent.

Figure 3: “positive standard deviation”? The standard deviation would be always positive or zero but it would never be a negative value. Do you mean (Mean + SD)? If yes, (Mean – SD) would become negative, suggesting there are negative CHL concentrations at some regions where the shelf is narrow (e.g., 47 – 51o N and 56 – 58oN). Maybe the distribution is lognormal. If this is the case, it would be better to use the median and range (or 10 – 90 percentiles) in these plots. Please clarify this.

              For consistency, please use the “mg m-3”.

Figure 4: The unit of chlorophyll-a on the X-axis is incorrect.

           Table 2: I do not understand the purpose of providing this table. The table shows the performance of various products (not all of them are used in this study) in different study areas other than the British Columbia Coast. At least, I think this table should have only those studies which provide an assessment of three products, GC-CHL1, GC-Int, and GC-OC-CCI. Furthermore, studies that have not validated the chlorophyll products before using them in the analysis do not provide any substantial information. This section of the table should be removed and if needed these studies can be mentioned in the text.

Line 379: An extra space between “products,” and “there”.

Line 414-419: In lines 413 – 415, authors have acknowledged that using a 3 x 3 pixel averaged value from global chlorophyll products (with a spatial resolution of 16 km2) as a valid matchup for the in situ measurement would increase the disparity between measured and satellite-derived chl concentrations. Then what is the need for using a 3 x 3-pixel average, especially when the averaging is already done at the pixel level when L2 scenes are aggregated to make the L3 scene (or Global products)? Also, see my major comment for section 3.1.

Line 465: An extra space between “2018).” and “Here”.

Line 473 – 475: Pitarch et al. (2016) used the same stations to assess the performance of various global chlorophyll products in their study. I think this is a more valid approach when the goal is the comparison of the model/product performance of two or more candidates. See my major comment for section 3.1.

Line 521: “mg m-3

References:

I think this journal prefers numbers ([1], [2] etc.) instead of (xyz et al., 2010) in text. Please check this. Similarly, the format for citing the reference under “References” is different for this journal.

Reviewer 3 Report

The work is devoted to the problem, important from a practical point of view, of determining the best products of merged satellite ocean color data for estimating the chlorophyll-a concentration and checking the quality of the gap-free interpolated product. The work is suitable for publication in the Remote Sensing journal. However, additional major revision is required to improve the comprehensibility and reliability of the results obtained. The structure of the article also needs to be improved.

Comments, remarks, and suggestions:

1. L201. What is meant by the OCx algorithm when listing the various empirical algorithms used? I looked that this is a quote from the work (Jackson, 2020), but did not find an explanation there. Usually, OCx is just the family of algorithms OC2, OC3, OC4, etc., see https://oceancolor.gsfc.nasa.gov/atbd/chlor_a/. Here it is desirable to give an exact list of algorithms (without specifying the scanner). Perhaps it is contained in the metadata of the data arrays used, or in other descriptions. This will allow you and the article readers to better interpret and understand the results.

2. L211-L212. Why did you use these time window values to compare the data? Is this an intuitive choice or the result of an analysis? For example, the criterion could be a significant decrease in the quality of the validation as the time window reaches some values.

3. Formula (1). Do I understand correctly that a filtered mean of the values of the surrounding pixels was used for comparative analysis? This is implied, but not explicitly written. Please add this information.

4. L216. You calculate the median in the sample, but it seems that these values ​​are not used anywhere else in the article, neither for analysis nor for quality criteria. You must either indicate how you used the median value or not mention it at all.

5.L234. Please specify for which specific metrics you log-transformed the data: MdAD - yes, BIAS - yes, r2 - yes, slope - yes, RMSE - no?

6. In your analysis, you use r-value and r-squared value. Please explain in methods what r-value is. Moreover, your r-value square is not equal to the r-squared value. You either have log-transformed or non-log-transformed data, or non-linear regression is used. Please clarify this point so that the reader does not have to guess and it is easier to interpret your results.

7. Item 3.2. It is not entirely clear why this part of the article is needed in accordance with its title, if the use of a single satellite radiometer with a good duration of operation (for example, MODIS-Aqua) can provide a similar result. Let us assume that this is an additional validation in terms of the correctness of the spatio-temporal distribution of the chlorophyll-a concentration. But then I would add here the tasks for which the use of merged products is more suitable: for example, determining the time of the spring and autumn algae bloom, or plot map for some selected short time period with an interesting natural process, or analyzing climate trends for the maximum possible period due to data merging. Thus, a comparison with known data also could be made. And for seasonal climatology, a single long-working ocean color radiometer, for example, MODIS-Aqua, is also perfect. For interest, you can compare the seasonal climatology built from the data of one radiometer (MODIS-Aqua) and the merged product. Will the merged product bring any new features? Do any artifacts appear?

So, clarify the purpose of this item in more detail and select more illustrative examples of the use of merged products.

8. Item 3.3, L330. It is necessary to make the structure of the article more standard and provide figures as they appear in the text, and table 2 should be moved to the discussion.

9. Table 1. Additionally, it is necessary to make the same analysis for the same samples so that there are exact matches in the number of points used for OC-CCI, GC, and GC int. This will make it possible to review the possible probability that in the case of OC-CCI additional points are used that worsen the result. And besides this, it will be interesting to compare GC int obtained from OC-like algorithms and GC obtained from the GSM algorithm. In general, using the current form of table 1 and the requested additional table will make it possible to conduct a full analysis.

10. Table 1. Your MdAD is the median estimate, which is more robust to outliers. And RMSE is an average, and even a quadratic estimate, i.e. strongly influenced by outliers. Therefore, comparing the RMSE result for the OC-CCI algorithm with the GC algorithm in the "case 2" waters can have a special meaning. I can assume that this is because OC-CCI had fewer outliers compared to GC in the "case 2" waters (or at least their number was similar). You give some discussion in general terms in the "discussion" section but without specific examples.

11. Discussion. Describe table 2 in the text with introductory sentences, otherwise, it suddenly appears during the discussion. It is difficult for me to check whether the metrics were calculated in the same way in your article and the cited works in table 2. Is the data log-transformed? Сalculation of regression parameters or comparison of two data vectors is used?  Etc. I hope you have made such an agreement. And if yes, then indicate it in the text.

12. You have found that GC is better than OC-CCI. At the same time, the first one was obtained by the quasi-analytical GSM model, and the second by empirical models. You also say that the version of the GC obtained by the empirical OS5 / CI models turned out to be very bad and you did not even begin to consider it, although GC int is also obtained by the empirical OC-like algorithms OC3 and OC5. There are also problems in applying algorithms using 412 or 443 nm channels in case of poor atmospheric correlation of satellite data. In general, all these questions about empirical and quasi-analytical algorithms are very interesting and deserve a separate discussion.

13. In general, in the discussion you have presented a rather dense text that is difficult to analyze. I recommend breaking it down into sections to make it easier to read and understand your results.

14. I recommend adding brief recommendations on the use of each of the analyzed merged datasets to the abstract.